# Electrophysiologic Effects of Growth Hormone Post-Myocardial Infarction

**DOI:** 10.3390/ijms21030918

**Published:** 2020-01-30

**Authors:** Konstantinos V. Stamatis, Marianthi Kontonika, Evangelos P. Daskalopoulos, Theofilos M. Kolettis

**Affiliations:** 11st Department of Cardiology, University Hospital of Ioannina, 45500 Ioannina, Greece; kwstas.stamatis100@gmail.com; 2Cardiovascular Research Institute, 45500 Ioannina, Greece; markontonika@yahoo.gr; 3Pole of Cardiovascular Research, Institut de Recherche Expérimentale et Clinique, Université Catholique de Louvain, 1200 Brussels, Belgium; evangelos-panagiotis.daskalopoulos@uclouvain.be

**Keywords:** growth hormone, myocardial infarction, ventricular tachyarrhythmias, structural remodeling, electrophysiologic remodeling

## Abstract

Myocardial infarction remains a major health-related problem with significant acute and long-term consequences. Acute coronary occlusion results in marked electrophysiologic alterations that can induce ventricular tachyarrhythmias such as ventricular tachycardia or ventricular fibrillation, often heralding sudden cardiac death. During the infarct-healing stage, hemodynamic and structural changes can lead to left ventricular dilatation and dysfunction, whereas the accompanying fibrosis forms the substrate for re-entrant circuits that can sustain ventricular tachyarrhythmias. A substantial proportion of such patients present clinically with overt heart failure, a common disease-entity associated with high morbidity and mortality. Several lines of evidence point toward a key role of the growth hormone/insulin-like growth factor-1 axis in the pathophysiology of post-infarction structural and electrophysiologic remodeling. Based on this rationale, experimental studies in animal models have demonstrated attenuated dilatation and improved systolic function after growth hormone administration. In addition to ameliorating wall-stress and preserving the peri-infarct myocardium, antiarrhythmic actions were also evident after such treatment, but the precise underlying mechanisms remain poorly understood. The present article summarizes the acute and chronic actions of systemic and local growth hormone administration in the post-infarction setting, placing emphasis on the electrophysiologic effects. Experimental and clinical data are reviewed, and hypotheses on potential mechanisms of action are discussed. Such information may prove useful in formulating new research questions and designing new studies that are expected to increase the translational value of growth hormone therapy after acute myocardial infarction.

## 1. Introduction

Myocardial infarction (MI) remains a major health-related problem worldwide, despite major treatment advances such as the widespread use of prompt reperfusion strategies [1]. Acute coronary occlusion induces immediate biochemical alterations that impair the contractile performance of the ischemic myocardium and affect left ventricular (LV) hemodynamics. Moreover, acute MI is characterized by prominent changes in ventricular electrophysiology, often leading to life-threatening ventricular tachyarrhythmias (VTs), namely sustained ventricular tachycardia and ventricular fibrillation [1]. During the healing phase, dilatation of the infarcted and non-infarcted zones, along with the accompanying fibrosis, provide the substrate for re-entrant circuits, a common mechanism responsible for VTs [2]. Most patients with progressive LV enlargement and dysfunction present clinically with overt heart failure, an ominous disease-entity associated with high morbidity and mortality, including sudden cardiac death secondary to VTs. Driven by the high prevalence of coronary artery disease, ample research efforts have been devoted toward the prevention of acute-phase and long-term complications of MI. 

Growth hormone (GH), a 191-amino-acid single-chain peptide extracted from human pituitary glands, is abundantly expressed in the body, including the ventricular myocardium [3]. In fact, the GH receptor gene is expressed in the ventricular myocardium at a relatively higher extent, when compared to several other tissues [4]. GH is now considered a pleiotropic hormone exerting diverse actions, many of which are still under investigation [5]. Over the past two decades, beneficial effects of GH have been reported in animal models of MI, leading to an enhanced healing process and smaller infarct size, and, ultimately, to preserved LV size and function [6]. However, far less is known on the effects of such intervention on acute and chronic arrhythmogenesis, with most available information arising from only a few laboratories. The present article reviews the current state-of-the-art on the effects of GH on the LV myocardium during the acute, healing, and chronic phases of MI, placing emphasis on the electrophysiologic actions and the presumed underlying mechanisms. The actions of GH differ along the course of MI (Figure 1), with the potential benefits gradually diminishing in established congestive heart failure. In this article, the GH-effects will be examined separately for acute MI, the subsequent remodeling period, and chronic MI. 

## 2. Acute Myocardial Infarction

### 2.1. Primary Ventricular Tachyarrhythmias 

Acute interruption of blood supply to the ventricular myocardium inhibits oxidative metabolism, decreases cellular energy storages, and alters intra- and extra-cellular ion concentrations. These profound changes in the electrophysiologic milieu, occurring mainly in the region of injury, result in a wide spectrum of VTs. The reported incidence of primary VTs, a devastating complication of acute coronary occlusion, has varied widely in published registries, confounded by the inevitable delays in pre-hospital monitoring. Moreover, the post-mortem diagnosis of type-3 MI, caused by fatal VTs prior to medical attendance, requires demonstration of acute thrombosis in a coronary artery, raising occasional difficulties in its differentiation from other causes of sudden cardiac death [7]. Allowing for these limitations, the incidence of primary VTs complicating the early phase of acute coronary occlusion is estimated at 10% of patients [8]. The incidence of VTs during the subsequent phase of evolving myocardial necrosis has decreased in the present era of primary percutaneous interventions, which are widely applied as first-line treatment of acute MI [1]. However, such VTs, often referred to as phase-II arrhythmogenesis, are also important, despite their occurrence invariably in the hospital setting [9]. Thus, there is an urgent need for further investigation of the pathophysiology of ischemia-induced VTs, based on the associated increased morbidity and mortality during the acute and sub-acute MI-stages, with a view toward improving their management [10]. 

### 2.2. Growth Hormone Confers Antiarrhythmic Actions

GH [11] and its mediator, insulin-like growth factor-1 (IGF-1) [12], have been shown to exert cytoprotective actions on the injured myocardium, resulting in a smaller necrotic area. Based on the previously demonstrated correlation between arrhythmogenesis and the extent of the ischemic myocardium in animal models [13], the hypothesis has been put forward that GH may confer antiarrhythmic actions during acute MI. This hypothesis was explored in two experimental studies in rats [14,15] investigating the effects of pre-treatment with GH on the incidence of post-MI VTs. 

The first study, performed by our group, quantified VTs from continuous electrocardiography, obtained by means of implanted miniature telemetry transmitters [14]. These devices permit long-term recordings in conscious animals without the confounding effects of anesthesia, thereby enabling the assessment of early (phase-I) and delayed (phase-II) VTs in our protocol. We found that phase-I VTs were unaffected, but we observed lower arrhythmogenesis in rats pre-treated with GH during phase-II (Figure 2), resulting in lower arrhythmic and total mortality 24 h post-MI. These favorable results were subsequently confirmed by another group [15], reporting lower incidence of spontaneous and induced VTs by programmed electrical stimulation; the antiarrhythmic effect of GH was found comparable to that elicited by amiodarone, a potent agent, well-characterized in this regard [16]. 

### 2.3. Cytoprotective Effects 

The explanation for the antiarrhythmic actions of GH during acute MI remains controversial, and several factors may be operative [17]. Cytoprotective actions have been advocated after GH administration in various tissues [18]; there is evidence to support the view that these effects are also operative in the ischemic LV myocardium, based on the results of three experimental studies [19,20,21], all utilizing the rat MI-model. In the first [19], the size of the necrotic area was lower by 18% in animals treated with GH for two weeks; nonetheless, such measurements were performed in the chronic phase of MI, and may actually reflect ameliorated infarct expansion secondary to improved LV hemodynamics. In the second study [20], DNA in the cytoplasm was abundant in the controls, as shown by Feulgen-staining 3 h after MI-induction; in contrast, DNA-staining was scarce in the sub-endocardial regions after pre-treatment with GH, indicating intact nuclear membrane [20]. The optimal time-point for infarct size measurements in the rat MI-model is uncertain, albeit they are probably of highest value if made after completion of the necrosis wave-front [22]. Based on this rationale, we previously examined the effects of systemic GH administration on the size of the necrotic tissue using triphenyltetrazolium staining 24 h after permanent coronary ligation; we reported smaller infarct-size (by 24%) in treated rats when compared to the controls [21].

Several mechanisms may be operative in mediating the cytoprotective effects of GH. Activation of pro-survival pathways by GH is a strong candidate, an action exerted via Akt-phosphorylation, which is dependent on the activation of the phosphoinositide 3-kinase [18]. In turn, this process may decrease cardiomyocyte apoptosis, as previously demonstrated in rats after long-term GH-treatment post-MI [23]. Additionally, we found evidence of attenuated inflammatory response in the peri-infarct region in our experiments [21], strengthening previous conclusions on the anti-inflammatory properties of GH [24]. Indeed, inflammation is a critical component of myocardial injury, associated with the generation of reactive oxygen species, a process activated by toll-like receptor signaling, followed by upregulation of cytokines and chemokines [25].

### 2.4. GH and Myocardial Norepinephrine Content 

Despite the considerable evidence favoring the cytoprotective actions of GH on the ischemic myocardium, the finding of decreased infarct size could not be reproduced in our laboratory using an identical setting [14]; specifically, the infarct size in GH-treated rats was only lower by 12.5%, a difference that failed to reach statistical significance. Although this equivocal result can be attributed solely to statistical variance, it also raises the possibility of additional mechanisms underlying the observed antiarrhythmic action of GH during evolving myocardial necrosis. To shed further light into this caveat, we re-evaluated the incidence of VTs in a further study, using the ischemia/reperfusion protocol [26]. The rationale behind this protocol was that reperfusion, an intervention with well-proven effects on salvaging ischemic myocardial tissue [27], could unveil further antiarrhythmic actions of GH. 

In addition to ischemia/reperfusion, two important amendments should be pointed out in our protocol [26]: First, GH was not given as pre-treatment, but after reperfusion, thereby increasing the potential therapeutic relevance of the study. Second, GH was not given systemically, but was administered locally by intra-myocardial injections around the antero-lateral LV wall. We thought that such administration at the rim of the ischemic zone maximizes treatment-efficacy by targeting a highly arrhythmogenic myocardial area during acute MI; in addition, it facilitates the drawing of inferences on the underlying pathophysiologic mechanisms by focusing on local GH-effects. 

We measured the ischemic- and infarcted-zones 24 h post-MI using green-fluorescent microspheres and triphenyltetrazolium; the final size of myocardial necrosis (expressed either in reference to the area-at-risk or to the total LV area) did not differ between the GH-treated group and the controls [26]. Thus, it became apparent that the cytoprotective potential of GH was likely overshadowed by that elicited by reperfusion. Interestingly, however, the finding of lower incidence of VTs after GH-treatment persisted, as shown by the lower number of VT-episodes and their total duration during a 24 h-observation period [26]. This finding strongly suggests local GH-effects on the LV myocardium, accounting for the antiarrhythmic action, independent of cytoprotection. One such mechanism refers to the effects of GH on norepinephrine release from LV sympathetic nerve terminals, based on the well-established association between hypopituitarism and sympathetic hyperactivity [28]. Along these lines, muscle sympathetic nerve activity was shown to be markedly increased in hypopituitary patients than in normal subjects and correlated positively with diastolic blood pressure, whilst inversely correlating with serum IGF-I levels [29]. 

Significant evidence favoring the inhibitory effects of GH on norepinephrine release stems from an experimental study in rats, where GH-treatment early post-MI not only improved myocardial energy reserve, but also markedly decreased myocardial norepinephrine content and plasma levels [30]. These observations are in keeping with our monophasic action potential recordings from the border zone (between the infarcted and normal myocardium) 24 h post-MI; specifically, our control group displayed partially depolarized recordings with shortened action potential duration, indicative of local sympathetic stimulation, but resting potential and morphology were preserved in treated rats [14]. Moreover, non-invasive autonomic indices showed improved sympathetic activation after local intra-myocardial GH-administration in our ischemia/reperfusion protocol [26]. This finding of ameliorated sympathetic drive may reflect reduced norepinephrine spill-over, as demonstrated previously [30], and lends further support to the ‘norepinephrine hypothesis’, as discussed below. 

### 2.5. Local Norepinephrine Release During Acute Myocardial Infarction

The hitherto available evidence [14,15,26,30], examined collectively, is useful in formulating the hypothesis that GH may act locally on sympathetic nerve terminals, decreasing local norepinephrine release during acute MI (Figure 3). The latter process, occurring independently of central sympathetic activity, has long been recognized as a key element of arrhythmogenesis during acute ischemia and evolving MI [31]. Its importance is underscored by studies using various experimental models such as the ex vivo, Langendorff-perfused heart preparation or after in vivo sympathetic denervation, indicating critical involvement of local norepinephrine in myocardial necrosis and in the occurrence of VTs [32].

The ensuing electrophysiologic effects of high interstitial norepinephrine-levels are multifaceted, including elevated resting membrane potential, delayed after-depolarizations, and dispersion of repolarization that forms functional re-entrant circuits [2]. Adrenergic stimulation of the ischemic myocardium occurs initially via exocytotic norepinephrine release, a process partly inhibited by adenosine, which is rapidly accumulated in the myocardium immediately after acute coronary occlusion. With the progression of ischemia, local metabolic mechanisms become increasingly important, resulting in norepinephrine accumulation in the cytoplasm of the neuron. This is followed by norepinephrine transport from the neuron into the interstitial space, mediated by the reverse action of the specific reuptake-carrier [33]. As a result, there is an over 100-fold increase in norepinephrine concentrations within the extracellular space of the ischemic zone [33]. Such massive interstitial norepinephrine concentrations exert cytotoxic effects by generating superoxide anion free radicals and, mainly, by increasing metabolic demand [34]. In turn, these actions exaggerate myocardial ischemia and initiate a vicious cycle of further myocardial damage and LV dysfunction [31]. Based on the essential pathophysiologic role of catecholamine-induced myocardial damage, we feel that further research should be devoted to the investigation of the potentially beneficial effects of GH during acute MI. 

## 3. Structural Remodeling

### 3.1. Pathophysiology

Following acute coronary occlusion, regional myocardial necrosis elevates local wall stress, leading to the expansion of the non-contractile region. Infarct expansion early post-MI is intertwined with healing mechanisms, with their balance largely determining subsequent changes in non-infarcted regions [35]. In this regard, hemodynamic alterations in the infarct zone can trigger global topographic changes, collectively termed LV remodeling, characterized by LV dilatation and dysfunction. As a consequence, local wall-stress also rises in the non-infarcted myocardium, a process accompanied by fetal gene expression, activation of abnormal activation of signaling pathways, and altered calcium handling [36]. 

### 3.2. GH in Post-MI Remodeling 

The GH/IGF-I axis is activated within hours after acute coronary occlusion and participates in the inherent infarct-healing process; for example, a threefold increase in plasma GH levels was found in MI patients as early as 6 h after the onset of symptoms, rapidly declining three days thereafter [37]. Such activation is believed to participate in infarct healing, by evoking a hypertrophic response, which consists of augmented actin and myosin synthesis and increased cardiomyocyte volume [38]. Moreover, GH stimulates the synthesis of collagen-I and -III at the site of injury, leading to the formation of a firm scar [11]. 

The favorable effects of GH on the ischemic and non-ischemic myocardium may have translational value in the post-MI setting, as indicated by a number of experimental studies using in vivo animal models. In a seminal work, Cittadini et al. [39] demonstrated that GH induced hypertrophy of the non-infarcted myocardium of rats, evidenced by echocardiography and morphometric histology, resulting in attenuated LV dilatation and improved LV systolic function. The same group subsequently reported beneficial GH-effects on LV hemodynamics and function in rats, associated with lower long-term mortality [23]. These results were reiterated by our group, after GH administration via a biomaterial-scaffold locally in the LV myocardium of rats [40]; three weeks post-MI, GH-treated animals displayed improved LV end-diastolic and end-systolic diameters, as well as more elegant remodeling-indices such as ventricular sphericity and wall tension index; moreover, microvascular density and myofibroblast count in the peri-infarct area were elevated after treatment (Figure 4). Thus, two further mechanisms became apparent that could explain the beneficial effects of GH on LV remodeling, with both, angiogenesis [41] and fibroblast-activation [42], thought to play an important role in stimulating inherent cardiac repair.

Our group has further examined the efficacy of a single dose of GH, administered selectively in the infarcted area via the intracoronary route in a large animal (porcine) model; we reported increased infarct thickness and enhanced angiogenesis in the peri-infarct area (Figure 5), leading to improved remodeling indices [43]. The intracoronary route for GH administration increases the translational value of this approach, rendering high clinical relevance in patients undergoing primary percutaneous interventions (PCI). Such early intervention aims at the (likely more attainable) prevention of LV remodeling, as opposed to its reversal at subsequent stages of overt heart failure. This concept was subsequently confirmed in a similar large animal model [19] after a single dosage of IGF-1 [44]. The intracoronary approach surfaced later into a small-scale clinical trial performed by the same investigators in patients with acute MI undergoing PCI [45]. The study randomized 47 such patients with depressed LV function after successful PCI into three groups, namely high-dose IGF-1, low-dose IGF-1, and saline, in addition to optimal medical therapy. No safety concerns were raised during or after intracoronary administration. Two months post-MI, remodeling-indices were improved after high-dosage IGF-1, as shown by smaller LV dimensions and mass, as well as higher stroke volume; there was also evidence of decreased fibrosis examined by late gadolinium enhancement in magnetic resonance imaging [45]. We feel that these very encouraging results call for larger scale clinical trials in acute MI.

## 4. Electrophysiologic Remodeling

### 4.1. Pathophysiology

In addition to structural LV alterations, post-MI remodeling is also associated with abnormal electrophysiologic properties, with the resultant arrhythmogenesis accounting for approximately 50% of mortality in such patients [2]. At the cardiomyocyte level, electrophysiologic remodeling affects calcium handling and adrenergic signaling, whereas prolonged action potential duration often elicits early after-depolarizations that can trigger VTs. Moreover, healed MI is characterized by areas around the scarred tissue, displaying marked variability in electrical conduction, and refractoriness that can sustain re-entrant circuits. 

### 4.2. Growth Hormone and Electrophysiologic Remodeling

In contrast to the available information on structural LV alterations, very little is known on the effects of GH on arrhythmogenesis during the sub-acute and chronic MI-phases. Attempting to bridge this gap, our group investigated indices of electrophysiologic remodeling after intra-myocardial GH administration in rats [46]; the assessment was performed two weeks post-MI, by which time the remodeling process was considered complete [47]. We included a wide range of electrophysiologic parameters such as programmed electrical stimulation and monophasic action potentials, coupled with the evaluation of electrical conduction and repolarization by means of a multi-electrode array. We found that treatment with GH preserved the shape and duration of the action potential at the infarct-border; moreover, local conduction was improved, evidenced by higher voltage-rise of the monophasic action potential and shorter activation-delay in multi-electrode recordings. Finally, repolarization-dispersion in the LV was markedly lower in GH-treated rats, which, in fact, displayed values similar to those seen in sham-operated animals [46]. Interestingly, we also observed an improvement in the right ventricular (RV) electrophysiologic parameters. As a result of these effects, the incidence of induced VTs after programmed electrical stimulation was lower in GH-treated rats, when compared to the controls [46]. Several mechanisms may underlie these favorable effects of long-term treatment with GH on the electrophysiologic remodeling, as discussed below.

### 4.3. Hemodynamic Improvement

GH has been shown to exert positive inotropic effects in animal models of heart failure [48]. The underlying mechanisms for this action remain elusive, with several mechanisms proposed such as increased calcium-sensitivity of the contractile proteins [49], and augmented calcium systolic release from the sarcoplasmic reticulum, mediated by increased density of ryanodine receptors [50]. Based on the correlation between hemodynamic disturbances in the failing heart and changes in action potential duration, it can be postulated that enhanced LV systolic function after GH-treatment may have improved the electrophysiologic milieu. The differences in RV electrophysiologic parameters in our study support this assumption, in keeping with the well-described hemodynamic and electrophysiologic RV-alterations that accompany post-MI LV remodeling [36].

### 4.4. Attenuated Fibrotic Response in the Remote Myocardium

Fibrosis in the non-infarcted myocardium accompanies LV remodeling and is considered the hallmark of arrhythmogenesis during the remodeling period and the chronic phase [2]. There is evidence to support the view that GH administration improves the fibrotic response in the non-infarcted myocardium, originating mainly from two studies [23,51]. Specifically, extracellular matrix remodeling was improved in GH-treated rats post-MI, as shown by preserved collagen-framework and uniform deposition of collagen-I fibers, a pattern contrasting the prominent disarray observed in the control group [23]. These results are in line with the absence of collagen deposition in the LV myocardium of normal rats, following long-term GH administration [52]. Moreover, these findings confirm earlier observations of a markedly reduced amount of collagen-I and fibronectin in the non-infarcted myocardium of rats after chronic GH-treatment [51]. Therefore, it appears that GH has the potential to attenuate the fibrotic response in the remote, non-infarcted LV myocardium [53] (Figure 6). 

### 4.5. Preservation of the Peri-Infarct-Area

The anti-fibrotic effects of GH in the remote, non-infarcted myocardium may seem contradictory to the well-described wound healing GH properties exerted by fibroblast activation; along these lines, we have reported a six-fold higher myofibroblast count after chronic GH administration, but, importantly, these were confined into the infarct and peri-infarct areas [40]. Such fibroblast activation in areas adjacent to the infarct-scar is considered beneficial and likely contributes to ameliorated infarct expansion [54]. The electrophysiologic consequences of activated cardiac fibroblasts and myofibroblasts are largely unknown; although these cells were initially believed to be electrically inert, electrophysiologic properties are now increasingly recognized, with both, proarrhythmic and antiarrhythmic effects reported after myocardial injury [55]. Thus, further work is needed on the electrophysiologic characterization of myofibroblasts post-MI and the possible effects of GH, which will allow more accurate conclusions on medium-term arrhythmogenesis.

An interesting finding after short-term GH administration in our experiments [21] was the increased mRNA synthesis of Fas, a cell-surface protein belonging to the tumor necrosis factor receptor super-family. This finding was prominent in the myocardium surrounding the infarcted tissue, and has been previously linked with the loading conditions in the LV [56]. Contrasting its pro-apoptotic role in other cell types, Fas does not seem to induce apoptosis in cardiomyocytes, but, instead, induces a hypertrophic response by glycogen synthase kinase 3β inhibition and concurrent Akt/protein kinase B activation [57] (Figure 7). Therefore, this process may actually confer beneficial effects under conditions of elevated wall-stress locally in the border-zone myocardium [58]. Taken together, it seems that GH ameliorates wall-stress and infarct expansion early post-MI, and thereby preserves the peri-infarct myocardium and decreases the substrate for re-entrant circuits that can sustain VTs. 

### 4.6. Differential Growth Hormone-Effects on Fibrosis

GH treatment post-MI appears to elicit two distinct effects, namely enhanced healing in the infarcted [11] and surrounding myocardium [21,40], along with attenuated fibrotic response in the non-infarcted myocardium [23,51]. Although the activation of fibroblasts selectively in the infarct and peri-infarct areas explains the former action, the mechanisms underlying the latter remain speculative, with two potential explanations having been brought forward [53,59]. The first refers to apoptosis in the non-infarcted myocardium, with the ensuing replacement fibrosis caused by residual interstitial edema, characterized by high fibrinogen content [60]. Based on the anti-apoptotic properties of GH, the prevention of subsequent interstitial edema may eventually attenuate fibrosis and, thereby, structural remodeling of the non-infarct zone [53]. A second, perhaps more likely, explanation for the absence of remote fibrosis refers to the actions of GH on transforming growth factor-β (TGF-β), a crucial regulator of fibrosis. Specifically, extracellular matrix and LV systolic and diastolic function remained normal after chronic GH-excess in rats, in contrast to the pattern observed after low-dosage angiotensin-II administration. Moreover, the expression of plasminogen activator inhibitor-1, as a marker of fibrosis caused by TGF-β, was increased after angiotensin II, but only marginally after GH; of note, a similar pattern was seen in fibronectin, collagen-I, and collagen-III (Figure 8). These findings indicate that the effects were mediated by a direct action of GH, causing suppression of TGF-β signaling via de-phosphorylation of p38 MAPK [59]. Based on these data, it seems that an important action of GH in the post-MI setting lies within the prevention of fibrosis in the remote, non-infarcted myocardium [53]. 

## 5. Chronic Phase Clinical Trials

In a substantial proportion of post-MI patients, LV remodeling progresses into overt heart failure, particularly in those with extensive anterior infarcts. The prevalence of heart failure is increasing, with its ominous prognosis posing a major health-related problem worldwide due to the epidemiology of coronary artery disease. Driven by its beneficial effects in animal studies, GH was examined as a potential adjunctive therapy in patients with remote MI and HF. Perhaps surprisingly, only two very small clinical studies have been reported in such patients [61,62]. The first was non-randomized, and included seven patients (55 ± 9 years of age) with congestive heart failure and LV dysfunction; they were assessed at the baseline, three months after GH treatment, and three months after cessation of treatment, whereas a control group was absent. GH improved clinical symptoms and exercise capacity as well as LV hemodynamic parameters such as pulmonary capillary wedge pressure and cardiac output. Moreover, posterior wall LV-thickness increased and end-diastolic and end-systolic volume decreased after treatment, although the LV ejection fraction remained stable [61]. The second study randomized patients with mild to moderate heart failure into GH and control; there were eight patients with previous MI, together with 13 patients with dilated cardiomyopathy and one patient with valvular heart disease. After three months, no differences were found (between active treatment and the control) in functional capacity or LV function, assessed by radionuclide angiography and Doppler echocardiography; it should be noted that no information is given on the outcome in reference to the etiology of heart failure [62]. Based on the results of this trial, GH therapy in the post-MI setting was disfavored, especially after the neutral results of another randomized trial examining patients with dilated cardiomyopathy [63]. 

## 6. Future Directions

Clinical interest in GH post-MI has recently resurfaced after the publication of two small-scale clinical trials [64,65] that led to an ongoing, observational, multicenter registry (NCT02335801). Expanding on this trend, we believe that there is a need for designing clinical trials that will examine the effects of GH, targeting specific patient populations; indeed, major pathophysiologic differences exist not only between patients with dilated cardiomyopathy and coronary artery disease, but also among different stages of MI. In this respect, the timing of GH administration appears of paramount importance, with administration early post-MI aiming at preventing adverse LV structural and electrophysiologic remodeling. Local GH delivery directly into the myocardium is expected to maximize efficacy, whilst avoiding potential untoward effects. The intracoronary root is attractive in patients undergoing percutaneous interventions [6], whereas local administration via the epicardium confers the advantage of sustained action in patients undergoing heart surgery, especially when GH scaffolds are used, enabling controlled-release [66]. In the remaining patients, short-term systemic administration presents a valid alternative (Figure 9). 

Patients with remote MI and congestive heart failure present complex neuro-hormonal responses including enhanced sympathetic drive and activation of the renin–angiotensin–aldosterone system. However, the beneficial effects of GH may still ensue, even in patients with advanced heart failure, after careful selection that will take into account multiple metabolic parameters [64,65]. 

## 7. Conclusions

Following acute coronary occlusion, GH exerts cytoprotective actions and attenuates local myocardial norepinephrine release, thereby limiting infarct size and lowering arrhythmogenesis. During subsequent stages, GH facilitates infarct healing and decreases infarct expansion, leading to improved local LV hemodynamics. The preserved electrophysiologic properties of the peri-infarct area, along with the prevention of fibrosis in the remote, non-infarcted myocardium, attenuate adverse structural LV remodeling and arrhythmogenic substrate formation. Multiple mechanisms are likely operative, many of which remain incompletely understood, necessitating further research on this promising approach in the post-MI setting. The conclusions derived from in vitro and in vivo experiments could be tested in clinical trials, targeting specific patient populations and end-points. 

## Figures and Tables

**Figure 1 ijms-21-00918-f001:**
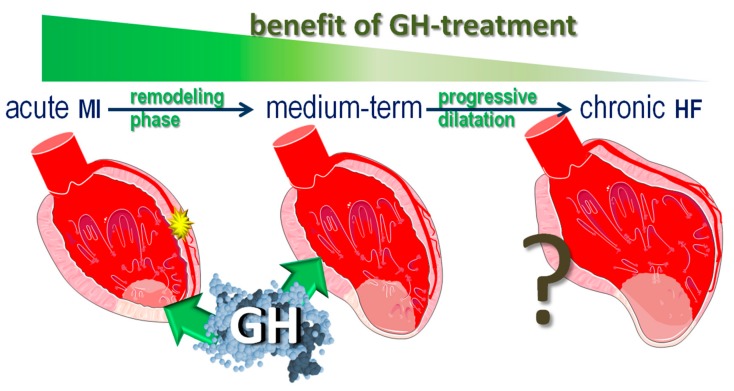
Growth hormone (GH) in myocardial infarction. The actions of GH differ along the course of myocardial infarction. It appears that the potential benefit of GH-treatment diminishes over time, pointing toward treatment strategies that aim to prevent adverse remodeling.

**Figure 2 ijms-21-00918-f002:**
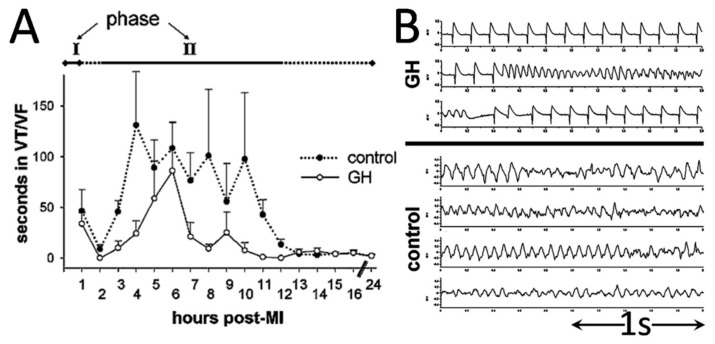
Ventricular tachyarrhythmias during acute infarction. (**A**) The total duration of tachyarrhythmias was shorter in the GH-treated group, mainly during phase II. Representative examples are shown in panel (**B**) From Elaiopoulos et al. Clin. Sci. (Lond), 112 (2007) 385–391 [12], with permission.

**Figure 3 ijms-21-00918-f003:**
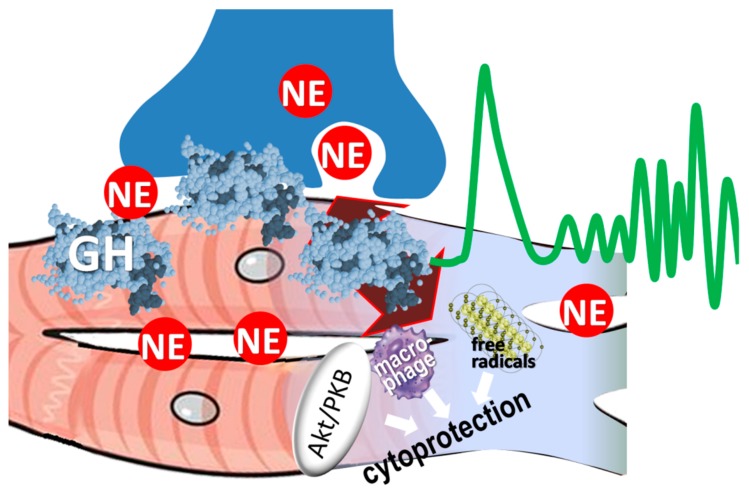
Proposed mechanisms of antiarrhythmic effects of GH. Two mechanisms may explain the antiarrhythmic effects of growth hormone, namely cytoprotection and reduced norepinephrine (NE) interstitial content.

**Figure 4 ijms-21-00918-f004:**
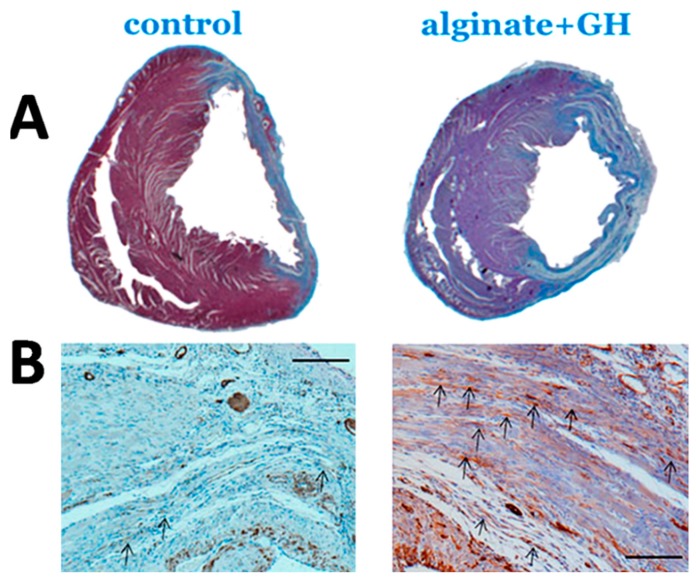
Structural remodeling. (**A**) Infarct thickness (Heidenhain’s AZAN-trichrome staining, scale: 500 mm) was preserved after treatment with growth hormone (GH), administered via an alginate scaffold. (**B**) Myofibroblast density in the peri-infarct-area (α-smooth-muscle-actin staining, arrows, scale: 50 mm) was higher after treatment. Adapted from Daskalopoulos et al., Growth Factors 33 (2015) 250–258 [37], with permission.

**Figure 5 ijms-21-00918-f005:**
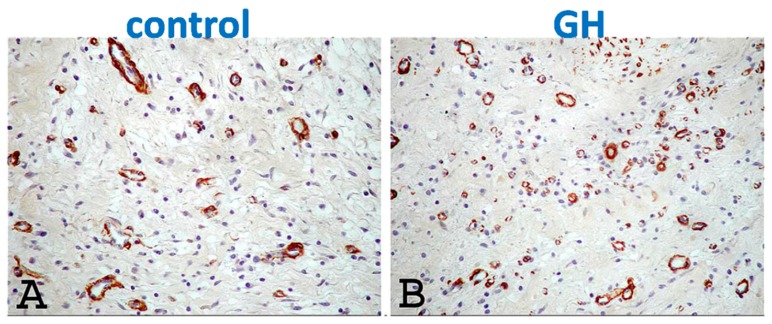
Angiogenesis. Angiogenesis in the peri-infarct area was enhanced after intra-coronary growth hormone (GH) (**B**), compared to controls (**A**). From Mitsi et al., Growth Horm IGF Res. 16 (2006) 93–100 [40], with permission.

**Figure 6 ijms-21-00918-f006:**
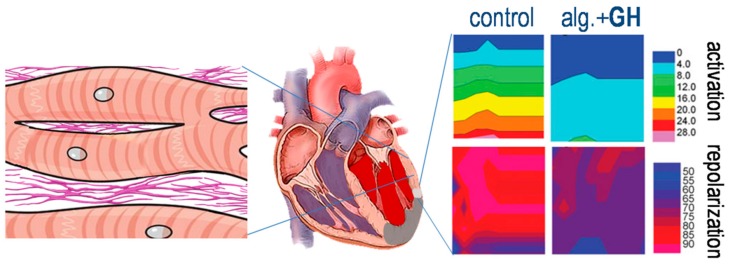
Electrophysiologic remodeling. Two proposed mechanisms of ameliorated electrophysiologic remodeling after treatment with growth hormone (GH), administered via an alginate (alg.) scaffold: prevention of fibrosis in the non-infarcted myocardium (**left** panel), and preservation of the peri-infarct area (**right** panel), resulting in improved conduction and repolarization indices. Containing data from Kontonika et al., Growth Factors 35 (2017) 1–11 [43], with permission.

**Figure 7 ijms-21-00918-f007:**
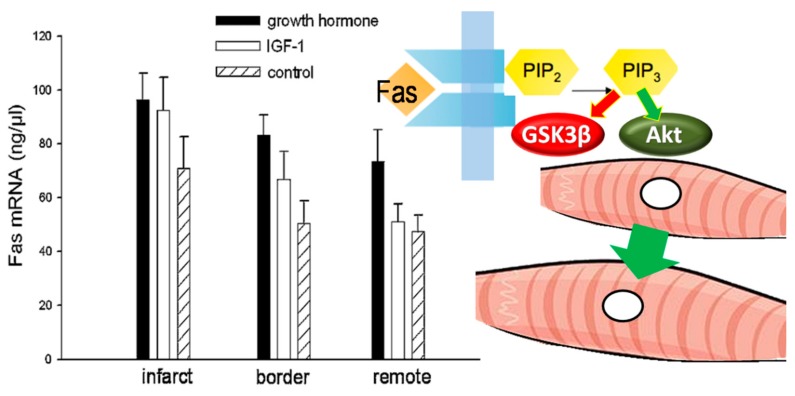
Cardiomyocyte hypertrophy. A possible mechanism of hypertrophy induced by growth hormone may be mediated by Fas, which inhibits glycogen synthase kinase 3β (GSK3β) and activates Akt/protein kinase B. This process is dependent on the activation of the phosphoinositide 3-kinase (PIP_3_). Containing data from Hatzistergos et al., Growth Horm IGF Res 18 (2008) 157–165 [19], with permission.

**Figure 8 ijms-21-00918-f008:**
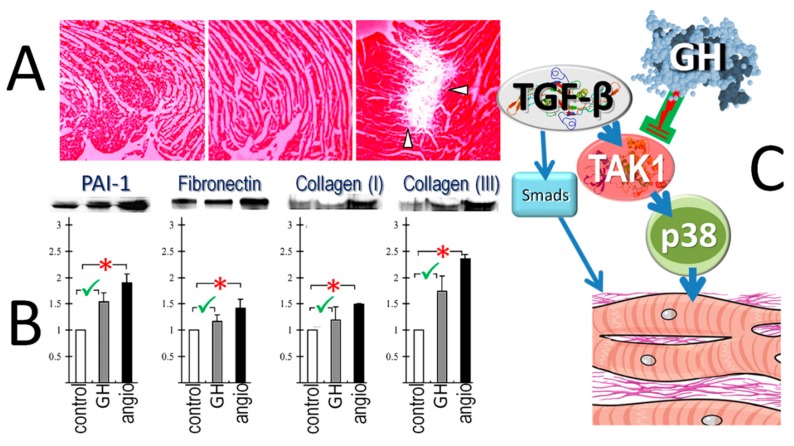
Growth hormone (GH) prevents fibrosis in the non-infarcted myocardium. GH inhibits transforming growth factor-β (TGF-β) signaling via TGF-β-activated kinase 1 (TAK1), leading to de-phosphorylation of p38 MAPK. As a result, the extracellular matrix is preserved, as shown by the absence of metalloproteinase activity (in situ zymography, (**A**). Furthermore, the expression of proteins related to TGF-β, such as plasminogen activator inhibitor-1 (PAI-1), fibronectin, collagen-I and collagen-III is reduced (**B**), and fibrosis is prevented in the remote myocardium (**C**). These effects contrast the fibrogenic effects of angiotensin-II (angio), pointed by arrows in (**A**) and by asterisks in (**B**). From Imanishi et al., Mol Cell Endocrinol (2004), 218, 137–146 [58], with permission.

**Figure 9 ijms-21-00918-f009:**
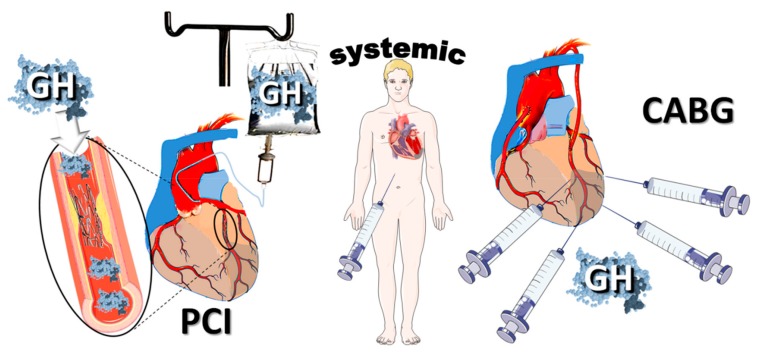
Routes for Growth hormone (GH) administration. GH could be administered via the intracoronary route in patients undergoing primary percutaneous interventions (PCI) for acute myocardial infarction. Direct intra-myocardial injections present an alternative route in patients undergoing coronary artery bypass grafting (CABG); it has the advantage of sustained action of GH via scaffolds enabling controlled release. Systemic administration is a valid option in the remaining patients.

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
