# Peer review of "Electrophysiologic Effects of Growth Hormone Post-Myocardial Infarction"

_ijms, 2020, doi:10.3390/ijms21030918_

Round 1
Reviewer 1 Report
In this work authors present an interesting and innovative topic on the implicated effects of growth hormone in the setting of post-myocardial infarction. This is a valid topic without much attention in the literature and so far showed modest translational value. Manuscript is generally well-written, although I do have some comments and concerns that I think would improve the current state of the manuscript:
Please change word "ramifications" in the Abstract with word "consequences". Third sentence - please also include ventricular tachyarrhyhmias such as ventricular tachycardia or ventricular fibrillation, not just VF. In the introduction define the acronym VT and then use it from there onwards. In the subsection on Acute myocardial infarction authors state that VF complicatrs about 10% of myocardial infarction cases. Is this true? There could be benefit on providing some up-to-date epidemiological data on this, also to differentiate this from sudden cardiac death or divide these two things into two different sentences, each with its own epidemiological data because these are separate entities and not synonyms. It is obvious that GH is pleiotropic and has differential effects in different stages of post-MI remodeling. However, authors wrote in section 4.3 that GH exerted positive inotropic in animal model of HF, however, earlier they go on to say that it blunts norepinephrine release locally in LV myocardium which definitely results with cardiodepressive effects. Are there any putative mechanisms possibly explaining these opposite effects of GH on cardiac inotropism? Similarly, in 4.4., authors go on to state that chronic GH administration in rats reduced collagen type I and fibronectin while earlier authors state that GH stimulates synthesis of collagen I and III at the site of injury, leading to the formation of a firm scar. As in line with previous point, are there any putative mechanism that could explain these very different actions of GH. Due to many effects of GH as elaborated in this manuscript, I think this manuscript would benefit from a summarizing figure that would show all relevant effects of GH in injured myocardium, and perhaps this figure could be didactically broken down in the acute phase, chronic phase, etc. so that the reader can understand these effects better. Authors did a good job in explaining these effects throughout the text, however, it is very easy to miss the broader context and get lost in many of the effects GH exhibits, especially among those that are mutually different in terms of pathophysiology and pharmacodynamic effects. I think that this paper lacks more insight on potential translational value of GH into clinical practice. Authors should dedicated a paragraph giving us a possible outlook of GH use in clinical practice. Do they anticipate that it could become a potential treatment solution, addition to current ACS/anti-remodeling armamentarium? If so, in their expert opinion, what would be the best timing and mode of potential and hypothetic GH administration? Just after MI? Later on? Intracoronary or intravenously? Please elaborate. It seems that GH interventions in humans showed modest results and not much applicability thus far. Why do authors think that there is more avenues to this worth further investigation and clinical testing in trials, etc.? I also think that the table summarizing most important preclinical and clinical studies (and broken down that way) and showing briefly main findings - effects of GH administration (as well as timing/mode of administration) would be beneficial for the readers to get a better summarizing picture of GH effects in trials and preclinical experiments thus far. What about growth hormone releasing hormone agonist in the context of myocardial infarction, such as in work of Kanashiro, PNAS 2010? Could you link GH to more upstream effects of GHRH in this setting so we can have a broader pathophysiological picture of this phenomenon.Author Response
Please see the attachment

Reviewer 2 Report
Dr. Konstantinos Stamatis and colleagues presented a timely comprehensive review on the role of growth hormone (GH) in post myocardial infarction.
There is growing evidence for possible role for GH in the treatment of congestive heart failure and myocardial infarction. GH/Insulin-like factor axis has been suggested as one of key players exerting cardioprotective effects, however exact mechanism (s) remain to be determined. The present review is well structured written summarizing mainly the in vivo work published by the authors and other investigators. Perhaps a separate section on existing/proposed clinicals studies within this specific area (if any) would add a value to this work. This review is of great value for academicians (electrophysiologists) and clinicians working in the CVD field.
Round 2
Reviewer 1 Report
The authors have answered all my comments effectively.
I have no further questions.